# Enhanced Performance of PVDF Composite Ultrafiltration Membrane via Degradation of Collagen-Modified Graphene Oxide

**Yonggang Hou** [1,2]**, Shenghua Lv** [1,]*****, **Haoyan Hu** [1]**, Xinming Wu** [2] **and Leipeng Liu** [1,]*****

[1] College of Bioresources Chemical and Materials Engineering, Shaanxi University of Science and Technology, Xi'an 710021, China; houyonggang@sohu.com (Y.H.); hyhy0609@outlook.com (H.H.)

[2] School of Materials Science and Chemical Engineering, Xi'an Technological University, Xi'an 710021, China; aimar_wu@xatu.edu.cn

***** Correspondence: lvsh@sust.edu.cn (S.L.); 4233@sust.edu.cn (L.L.); Tel.: +86-86168261 (S.L. & L.L.)

**Abstract:** The collagen obtained from chrome leather waste can be used to modify graphene oxide (GO) to prepare polyvinylidene fluoride (PVDF) composite ultrafiltration membranes, a process that is conducive to the recovery of leather waste, comprehensive utilization of GO and improved performance of the membrane. In this paper, collagen-modified GO (CGO) was prepared by degradation of collagen from chrome leather waste and used to prepare a PVDF composite ultrafiltration membrane. The results show that the carboxyl content of CGO and dispersion were improved. The water flux and flux recovery rate of the modified ultrafiltration membrane were improved. The bovine serum albumin (BSA) intercepted on the membrane surface was easy to clean and the antifouling performance improved. The performance of the membrane decreased when the GO content exceeded 0.75 wt%, while CGO can reach 1.0 wt% without agglomeration due to its good dispersion.

**Keywords:** PVDF; graphene oxide; ultrafiltration membrane; chrome leather waste; antifouling

## 1. Introduction

In the process of leather production, the utilization rate of raw leather is only 30% and the rest is discarded as scraps [1]. Chrome leather waste contains a lot of collagen protein that is usually considered a potential resource [2]. With the advantages of a high chromium removal rate by alkali degradation and less pollution by enzymatic hydrolysis [3], chrome leather waste can be efficiently and ecologically degraded into small-molecular-weight collagen for functional applications [4–6].

The structure of graphene oxide (GO) consists of hydroxyl, carboxyl, epoxy groups and other oxygen-containing functional groups [7,8]. Because of its good hydrophilicity and compatibility with polar materials, it is widely used in the preparation of composite materials [9,10]. However, GO tends to agglomerate when applied [11], which affects its application potential, thus it is necessary to improve its dispersion [12,13]. Functionalization of GO through carboxyl and other functional groups can improve its dispersion [14–16].

Polyvinylidene fluoride (PVDF) is widely used in filtration membrane materials due to its good mechanical properties and chemical stability [17,18]. However, due to its hydrophobicity, it becomes polluted easily, leading to decreased water flux in water treatment [19,20]. Improving the hydrophilicity of a PVDF membrane is an effective way to improve its antifouling performance [21]. Inorganic nanoparticles have small sizes and large specific surface areas, which can accelerate the mass transfer process of a membrane and facilitate the exchange of solvents and non-solvents, so they are used in PVDF ultrafiltration membranes to enhance their antifouling ability and water flux [22,23]. However, aggregation of nanoparticles on the polymer membrane will cause defects of the membrane structure, resulting in degradation of membrane performance [24]. By compounding with

GO, the aggregation of nanoparticles on the surface of the membrane can be improved, which will improve the performance of the membrane [25]. Through GO functionalization, the active functional groups make the casting solution form more voids in the film-forming process, which can adsorb pollutants and improve the antifouling performance.

In this paper, collagen obtained by two-step degradation of chrome leather waste was used to prepare modified GO (CGO) to improve its dispersion and PVDF composite ultrafiltration membrane was prepared using the CGO. The purpose was to recycle chromium-containing waste leather, improve the dispersion of GO and improve the water flux and antifouling properties of PVDF composite ultrafiltration membrane.

## 2. Experimental

### 2.1. Chemicals and Materials

All chemicals were analytical grade and used as received without further purification. The chemicals used were as follows: Potassium permanganate ($KMnO_4$), concentrated sulfuric acid ($H_2SO_4$, 98%), sodium hydroxide (NaOH), sodium nitrate ($NaNO_3$), hydrogen peroxide solution ($H_2O_2$, 30%), calcium oxide (CaO) and *N*-methyl pyrrolidone (NMP), which were procured from Tianjin Kemiou Chemical Reagent Co., Ltd., Tianjin, China, Other materials used in the experiment were as follows: Natural flake graphite (carbon content > 96%), chrome leather waste (water content < 10%), alkaline collagenase (200 $U \cdot mg^{-1}$), bovine serum albumin (BSA, 98%), polyvinylidene fluoride (PVDF, Mw = 250–450 kDa) and polyvinylpyrrolidone (PVP, K30), provided by Shaanxi Huaxing Experimental Technology Co., Ltd., Xi'an, China.

### 2.2. Preparation Process

#### 2.2.1. Degradation of Collagen-Modified Graphene Oxide

Calcium oxide, sodium hydroxide and alkaline collagenase were used to degrade waste leather scraps by a two-step process of alkali and enzyme degradation to prepare degraded collagen [26]. GO was prepared by the modified Hummers method [27]. The degraded collagen was added to the GO dispersion, reacted at 70 °C for 24 h, washed and freeze-dried to obtain degraded collagen-modified graphene oxide (CGO).

#### 2.2.2. PVDF Composite Ultrafiltration Membrane

PVDF composite ultrafiltration membrane was prepared with PVDF as the matrix, PVP as the pore-forming agent, NMP as an organic solvent and CGO as a hydrophilic additive.

GO or CGO was dissolved in NMP and underwent ultrasonic treatment for 4 h to form a uniform suspension. The casting solution was prepared according to Table 1 and stirred at 70 °C for 24 h to dissolve the PVDF and PVP powder in the NMP. The casting solution was placed in a vacuum oven at 70 °C for 12 h to remove bubbles. PVDF films were prepared by spin-coating at a speed of 500 $r \cdot min^{-1}$ for 10 s. After spin-coating, the films were heated at 60 °C in vacuum until the solvent was completely volatilized. Then, the residual NMP was washed with deionized water and the ultrafiltration membrane was stored in deionized water.

**Table 1.** Composition of casting liquid.

| No. | PVDF (wt%) | PVP (wt%) | GO (wt%) | CGO (wt%) | NMP (wt%) |
|-----|-----------|-----------|----------|-----------|-----------|
| M0 | 18 | 3 | 0 | 0 | 79 |
| M11 | 18 | 3 | 0.5 | 0 | 78.5 |
| M12 | 18 | 3 | 0.75 | 0 | 78.25 |
| M13 | 18 | 3 | 1.0 | 0 | 78 |
| M14 | 18 | 3 | 1.25 | 0 | 77.75 |
| M21 | 18 | 3 | 0 | 0.5 | 78.5 |
| M22 | 18 | 3 | 0 | 0.75 | 78.25 |
| M23 | 18 | 3 | 0 | 1.0 | 78 |
| M24 | 18 | 3 | 0 | 1.25 | 77.75 |

The preparation process of CGO/PVDF composite ultrafiltration membrane is shown in Figure 1.

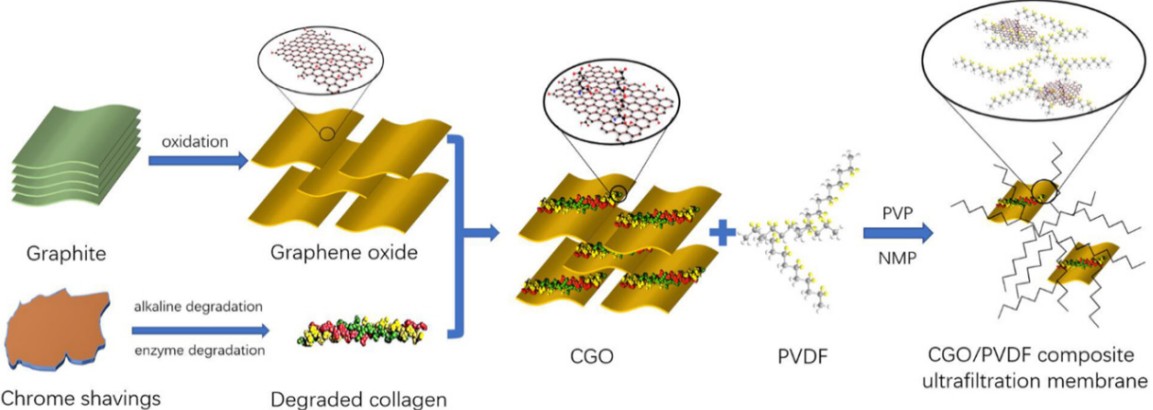

**Figure 1.** Diagram illustrating process of preparing CGO/PVDF composite ultrafiltration membrane.

### *2.3. Characterization and Testing*

2.3.1. Structural Characterization

1.　Chemical structure

Fourier-transform infrared spectroscopy (FTIR) spectra were recorded on a Vector-22 spectrometer (Bruker, Karlsruhe, Germany) in the range of 400 to 4000 cm$^{-1}$ with a resolution of 2 cm$^{-1}$. The samples were pressed with KBr into pellets before FTIR measurement. X-ray diffraction (XRD) was carried out using a D/max 2200 PC X-ray diffractometer (Rigaku, Tokyo, Japan) with Cu K$\alpha$ radiation ($\lambda$ = 0.15418 nm) at a scan rate of 5°·min$^{-1}$. X-ray photoelectron spectroscopy (XPS) was carried out using an AXIS Supra XPS spectrometer (Shimadzu, Kyoto, Japan) with an Al K$\alpha$ (1487 eV) X-ray source. Zeta potential was acquired with a Zetasizer Nano-ZS90 (Malvern, Malvern, UK).

2.　Micro structure

Scanning electron microscopy (SEM) images were obtained using an EVO 18 electron microscope (Zeiss, Jena, Germany) at 20 kV. The sample was sliced and transmission electron microscope (TEM) images were collected using a JEM-2010 (JEOL, Tokyo, Japan) at 200 kV.

3.　Pore structure

Porosity was calculated according to Equation (1):

$$\varepsilon = \frac{W_1 - W_2}{A \times l \times \rho_W}, \tag{1}$$

where $\varepsilon$ is the membrane porosity (%), $W_1$ is the dry membrane weight (g), $W_2$ is the wet membrane weight (g), $A$ is the membrane area (m$^2$), $l$ is the membrane thickness (m) and $\rho_W$ is water density (0.998 g·cm$^{-3}$).

The average pore size of the composite ultrafiltration membrane was calculated by pure water flux, thickness, effective area and so on. The calculation formula is the Guerout–Elford–Ferry Equation (Equation (2)) [28]:

$$r_M = \sqrt{\frac{(2.9 - 1.7\varepsilon) \times 8\eta l Q}{\varepsilon \times A \times \Delta P}}, \tag{2}$$

where $r_M$ is the average membrane pore size (m), $\varepsilon$ is the membrane porosity (%), $\eta$ is water viscosity (8.9 × 10$^{-4}$ Pa·s), $l$ is the membrane thickness (m), $Q$ is the water flux per

unit time ($m^3 \cdot s^{-1}$), $A$ is the membrane effective area ($m^2$) and $\Delta P$ is the operating pressure (0.1 MPa).

### 2.3.2. Water Contact Angle Testing

The water contact angle of the ultrafiltration membrane was measured by a DSA100 water contact angle analyzer. Each sample was measured at five locations and the average value was taken.

### 2.3.3. Testing for Pure Water Flux and Antifouling Performance of Membrane

All experiments were conducted at 25 °C with a filtration pressure of 0.1 MPa. For the measuring procedure, the pure water flux was recorded at 0.1 MPa every 3 min for 90 min and at least four replicates were collected to calculate an average value. The pure water flux of the membrane was calculated according to Equation (3):

$$J_W = \frac{V}{A \times t},$$
(3)

where $J_w$ is pure water flux ($L \cdot h^{-1} \cdot m^{-2}$), $V$ is the volume of pure water passing through the membrane per unit time (L), $A$ is the membrane area ($m^2$) and $t$ is the filtration time (h).

The solute rejection performance of the membranes was measured by using BSA as foulant. The concentration of the feed and permeation was evaluated by a UV-vis double-beam spectrophotometer at a wavelength of 280 nm. The percentage of solute rejection was calculated by Equation (4):

$$R = 1 - \frac{c_2}{c_1},$$
(4)

where $R$ is the BSA rejection rate of the membrane (%), $c_1$ is the concentration of the feed (1 mg·mL$^{-1}$) and $c_2$ is the concentration of permeation (mg·mL$^{-1}$).

The flux recovery rate (FRR) of the membrane was used to characterize the percentage of contaminated membrane reaching the initial level after cleaning, i.e., antifouling performance. The composite membrane with filtered BSA was rinsed with pure water, soaked in pure water for 10 min and then the pure water flux was measured again, which is the recovery flux. The FRR of the membrane was calculated according to Equation (5):

$$FRR = \frac{J_{rW}}{J_W} \times 100\%,$$
(5)

where FRR is the flux recovery rate of the membrane (%), $J_{rw}$ is the recovery flux of the membrane ($L \cdot h^{-1} \cdot m^{-2}$) and $J_w$ is the pure water flux of the membrane ($L \cdot h^{-1} \cdot m^{-2}$).

## 3. Results and Discussion

### 3.1. Structure Analysis

#### 3.1.1. Chemical Structure of GO and CGO

It can be seen from Figure 2a that the carboxyl stretching vibration peak of CGO near 1575 cm$^{-1}$ is stronger than that of GO and the peak shape is also widened, which indicates that the carboxyl content of CGO is higher than that of GO. For CGO, a new peak appeared near 3000 cm$^{-1}$, which is the stretching vibration peak of -CH$_2$, because the degraded collagen grafted with GO contains more -CH$_2$. In Figure 2b, the sharp peak at 10.7° for GO is its characteristic peak and this peak for CGO disappeared after modification. The sharp peak for CGO appearing at 26° was the characteristic XRD peak of graphene, indicating that the reaction of degraded collagen and GO led to a partial reduction of GO. At the same time, due to the reaction temperature of 70 °C, GO was also reduced at a higher reaction temperature. It can be seen from Figure 2c that GO peaks at 284.6, 285.6, 286.2 and 288.9 ev belong to C-C (C=C), C-N, C-O and O-C=O functional groups, respectively. Figure 2d shows that CGO peaks at 284.6, 285.6, 286.2, 287.4 and 289.3 ev belong to C-C (C=C), C-N, C-O, C=O and O-C=O functional groups, respectively. After being modified by degraded

collagen, the peptide chain of CGO still contained a C-N group due to its own structure, so the strength of C-N/C-NH$_2$ increased.

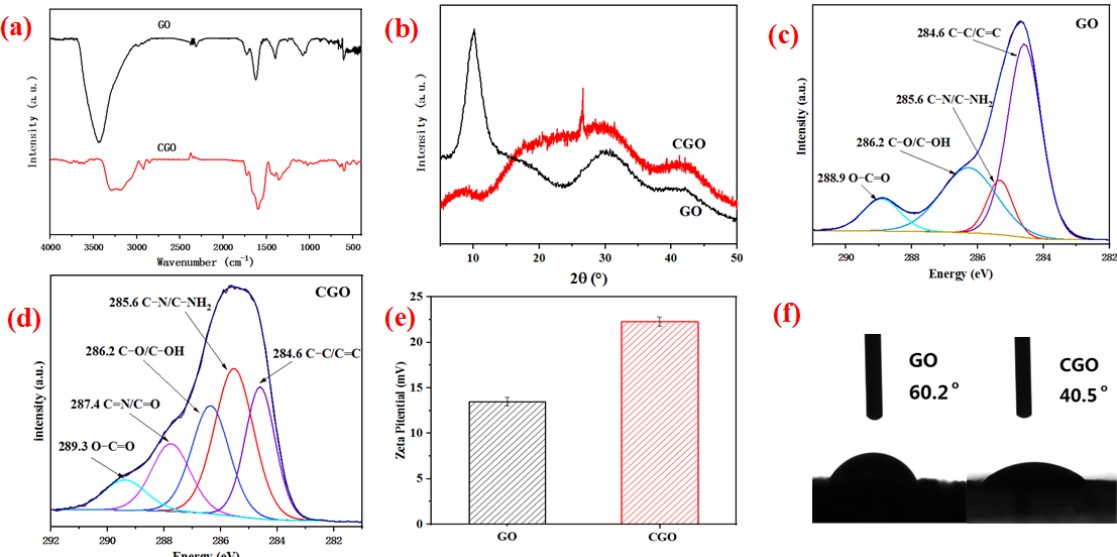

**Figure 2.** (**a**) FTIR of GO and CGO, (**b**) XRD patterns of GO and CGO, (**c**)XPS spectra of GO, (**d**) XPS spectra of CGO, (**e**) zeta potential and (**f**) water contact angle of GO and CGO.

The results of FTIR, XRD and XPS indicate that the degraded collagen was grafted with GO successfully and CGO contained more carboxyl groups. It can be seen from Figure 2e that the zeta potential of GO aqueous solution is −13.5 mv and that of CGO reaches −22.3 mv. Figure 2f shows that the contact angle of CGO was much lower than that of GO. Both showed that the carboxyl group of collagen gave CGO better hydrophilicity and increased its dispersion stability.

### 3.1.2. Micro Structure of PVDF Membrane

Figure 3a shows an SEM image of PVDF, where it can be seen that the PVDF has a smooth surface and compact texture. From the SEM image of GO/PVDF (0.75 wt%) in Figure 3b, it can be seen that GO of the slice layer is dispersed in the PVDF, but the dispersion is not uniform and GO, with a two-dimensional structure, will precipitate on the surface of the PVDF. As can be seen from the SEM image of CGO/PVDF (0.75 wt%) in Figure 3c, the CGO is coated by PVDF and there is no precipitated two-dimensional structural material of CGO on the surface of the sample.

In addition, a cross-sectional SEM image of CGO/PVDF is shown in Figure 3d, which demonstrates that the membrane is dense and compact and the CGO is uniformly embedded in the PVDF matrix. For further verification of the dispersion of CGO in PVDF, TEM images of CGO/PVDF are shown in Figure 3e. It can be seen from the images that the two-dimensional black lamellar material is CGO, the shadowy gray part is PVDF matrix and the CGO is relatively evenly dispersed in the PVDF, which is consistent with the SEM test results. Such a uniform and compact structure lays a foundation for the separation performance of water.

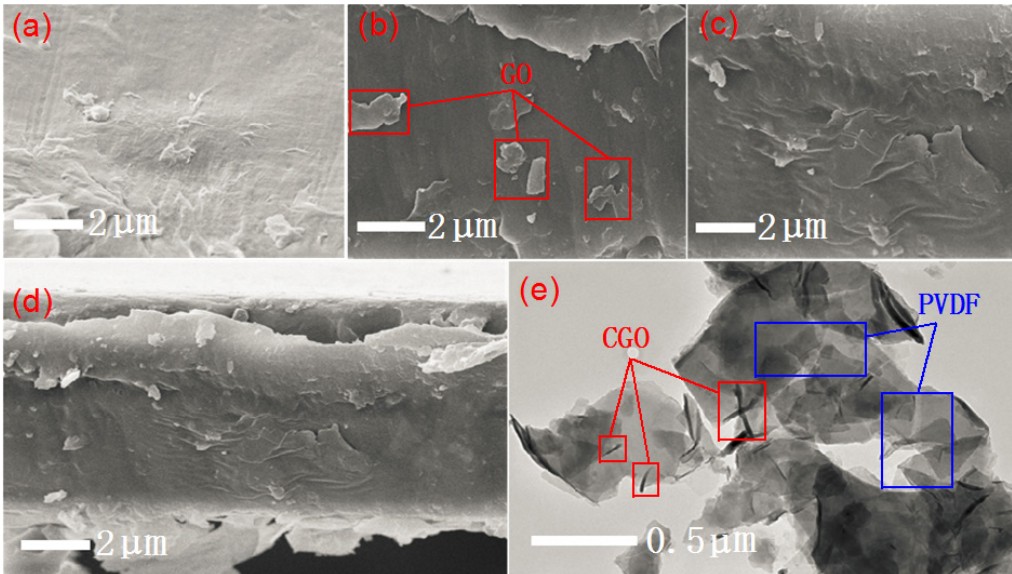

**Figure 3.** SEM images of (**a**) PVDF, (**b**) GO/PVDF and (**c**,**d**) CGO/PVDF. (**e**) TEM images of CGO/PVDF.

### 3.1.3. Pore Structure of PVDF Membrane

Figure 4a shows the porosity of the PVDF composite ultrafiltration membrane with different GO or CGO content. It can be seen that the porosity of the composite membrane was significantly higher with CGO than with GO and the highest porosity was 64.6%. Comparing the two membranes, it can be seen that maximum porosity of the composite ultrafiltration membrane with GO is reached when the GO content is 0.75 wt%, while for the membrane with CGO, the value is 1.0 wt%. This result reflects that CGO has more oxygen-containing groups than GO, so it has better dispersion in PVDF, so it can be added in higher amounts without agglomeration.

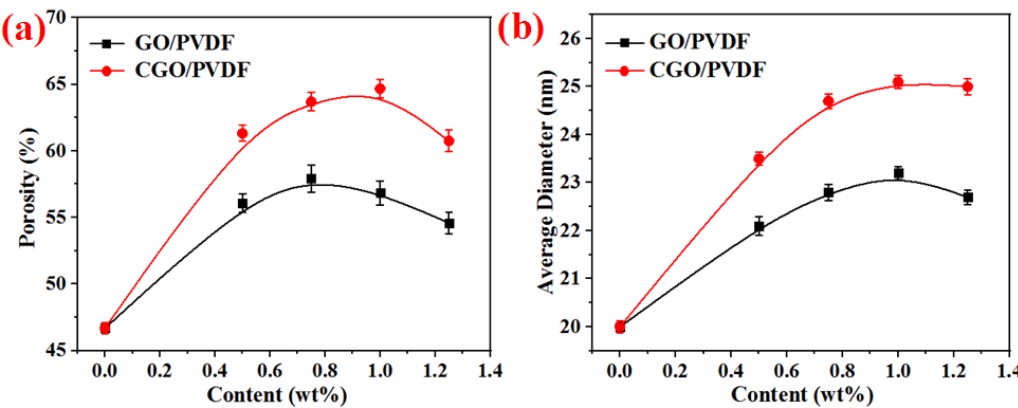

**Figure 4.** (**a**) Porosity and (**b**) average diameter of PVDF composite ultrafiltration membrane.

Figure 4b shows the relationship between the average pore size of PVDF composite ultrafiltration membrane and the amount of GO or CGO. It can be seen that the average pore size first increased and then decreased. The maximum was reached with the addition of 1.0 wt% GO and CGO, then it decreased. The reason for this result is similar to that for the effect of the added amount on porosity. In addition, when the amount of GO and CGO is too large, they will agglomerate, resulting in decreased pore size. The average pore size of the ultrafiltration membrane with GO increased by about 3 nm, while that with CGO increased by about 5 nm. This is due to more active groups in the CGO structure. Comparing the pore size of the two samples, it can be seen that after reaching the maximum

value, the pore size of the membrane with GO decreased rapidly, while that with CGO decreased relatively slowly. The reason is that when the content of GO is too high, it cannot be evenly dispersed in the PVDF matrix and will agglomerate. However, the pore size of the composite membrane containing CGO decreased slowly, indicating that CGO had better dispersion in the matrix than GO. This is due to the fact that CGO contains more active functional groups and is does not agglomerate easily.

### 3.2. Performance of PVDF Composite Ultrafiltration Membrane

Figure 5 shows water fluxes, BSA rejection rates, flux recovery rates and a schematic diagram of PVDF, GO/PVDF and CGO/PVDF ultrafiltration membranes. The water flux of ultrafiltration membrane is an important parameter to characterize membrane permeability [29]. Figure 5a shows the water flux of PVDF, GO/PVDF and CGO/PVDF ultrafiltration membranes. It can be seen that the water flux of the modified composite ultrafiltration membrane is higher than that of the pure PVDF ultrafiltration membrane. The water flux of the composite membrane with GO is increased by about 20 $L \cdot m^{-2} \cdot h^{-1}$, while that of the membrane with CGO is increased by nearly 40 $L \cdot m^{-2} \cdot h^{-1}$ and both show the trend of first increasing and then decreasing. When the GO dosage was 0.75 wt%, the water flux reached the maximum, which was 68 $L \cdot m^{-2} \cdot h^{-1}$. When the GO dosage was 1.0 wt%, the water flux decreased to 67 $L \cdot m^{-2} \cdot h^{-1}$, indicating that the maximum dosage was between 0.75 and 1.0 wt%. The water flux of the composite membrane with CGO reached the maximum at 1.0 wt%, which was 92 $L \cdot m^{-2} \cdot h^{-1}$ and then decreased.

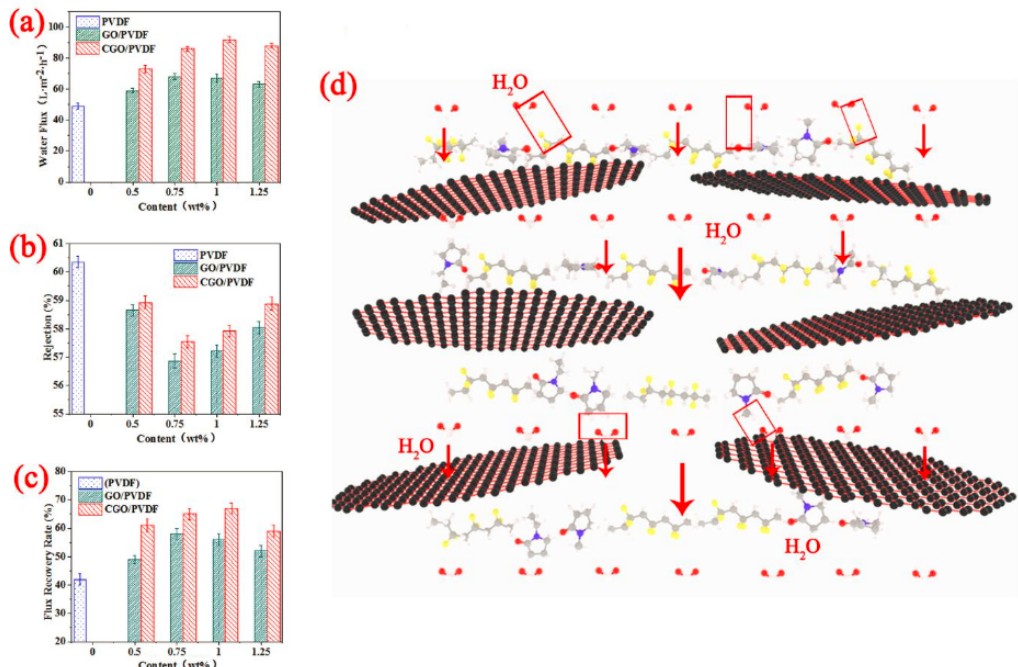

**Figure 5.** (**a**) Water fluxes, (**b**) BSA rejection rates, (**c**) flux recovery rates and (**d**) a schematic diagram of a composite ultrafiltration membrane.

The reason for this result is that CGO has better hydrophilicity, which makes the composite membrane form more holes, which is conducive for more water to pass through. In addition, the hydrophilic groups on the membrane make it easier to adsorb water molecules, which means higher water molecule transfer efficiency in the membrane pores, so the water flux is greater. At the same time, the maximum addition of CGO is slightly larger than that of GO, which also reflects that CGO has better dispersion and can be more dispersed on PVDF substrate. The water flux of the two ultrafiltration membranes increased at first and then decreased, indicating that both GO and CGO had the tendency of aggregating when the dosage was too large, so that the hydrophilic groups exposed on

the surface of the membrane did not increase, but decreased instead. This is also a problem to be considered when GO is used as an additive for ultrafiltration membrane modification.

The rejection rate is an important parameter for characterizing the filtration performance of an ultrafiltration membrane [30]. In this paper, BSA was used to simulate the filtration performance of the ultrafiltration membrane for protein pollutants. It can be seen from Figure 5b that the rejection rate of BSA by the pure PVDF ultrafiltration membrane is 60.4%. With the addition of GO or CGO, the rejection rate decreases and both show a trend of first decreasing and then increasing. This is due to the increased porosity and average pore size of ultrafiltration membrane with the addition of GO or CGO. When the dosage of GO or CGO was 0.75 wt%, the rejection rate of the ultrafiltration membrane reached the minimum, which was 56.9 and 57.6%, respectively. However, with increased GO content, the GO agglomerates and porosity decreased, so the rejection rate increased. The data show that the rejection rate of the ultrafiltration membrane with CGO was higher than that with GO. This is because there are more active groups on the surface of the membrane with CGO added and some BSA molecules are intercepted by non-covalent bonds.

The flux recovery rate (FRR) is an important parameter to characterize antifouling performance [31]. Figure 5c shows the FRR of a PVDF ultrafiltration membrane with different GO or CGO additions. It can be seen that the FRR of a pure PVDF ultrafiltration membrane is about 42% and with the addition of GO or CGO, it first increases and then decreases. This result can be explained by hydrophilicity. The better the hydrophilicity of the composite membrane surface, the more easily water molecules can be adsorbed on the surface during the filtration process. Water molecules adsorbed on the surface will then form a hydration layer. This hydration layer helps to isolate the contact between the membrane surface and pollutants, such that the pollutants can be removed more easily and the FRR can be improved. Moreover, composite ultrafiltration membrane with CGO has more hydrophilic functional groups on the surface, which makes it easier to form a large hydration layer, so the FRR is higher. Comparing the FRR with the addition of GO or CGO, it was found that this trend was similar to water flux and porosity, which reached the maximum at 0.75 and 1.0 wt%, respectively.

Figure 5d shows a schematic diagram of the composite ultrafiltration membrane. As seen in the figure, the principle of preparing composite materials and the function of each component are as follows: Chrome leather scraps are waste, but they can be degraded to collagen by alkaline and enzyme degradation (this collagen may become a small molecular amino acid or peptide chain with multiple amino acids). The chemical structure of collagen contains functional groups similar to amino acids. The oxygen-containing functional groups on the graphene oxide structure include epoxy group, hydroxyl and carboxyl. CGO can be obtained by a ring-opening addition reaction between the amino group on collagen and the epoxy group on graphene oxide structure (or a reaction between the amino group on collagen and the carboxyl group on graphene oxide structure). CGO has a higher carboxyl content than GO, so it has better hydrophilicity. Because of the hydrophobicity of PVDF, its water flux is not high as that of a separation membrane. Improving the hydrophilicity of PVDF can improve its water flux. At the same time, it can also improve the elution of the membrane after separating protein pollutants, so as to improve the recycling efficiency. CGO was used as filler to form a composite material with PVDF in solvent NMP. PVDF is the matrix of the ultrafiltration membrane, PVP is the porogen, NMP is the organic solvent and CGO is the hydrophilic additive. The film-casting solution obtained by mixing these materials was heated to volatilize the organic solvent so as to form a membrane. After that, the CGO/PVDF composite ultrafiltration membrane was formed by putting the obtained membrane in water to remove NMP and porogen PVP. Compared with GO, CGO has higher carboxyl content and better hydrophilicity, so the water flux of the composite material formed with PVDF was also improved.

## 4. Conclusions

In this paper, alkaline degradation combined with enzyme degradation was used to degrade chrome leather waste to prepare degraded collagen, and an improved Hummers method was used to prepare GO. In order to improve the performance of the ultrafiltration membrane, CGO prepared by grafting degraded collagen with GO was added into PVDF. The results show that the carboxyl content of GO modified by degraded collagen was increased, which improved its dispersion and increased the amount of CGO added without agglomeration. The results show that the water flux and FRR of the composite ultrafiltration membrane increased with the addition of CGO. The BSA intercepted on the membrane surface was easy to clean and the antifouling performance improved. More CGO can be added to the composite membrane without agglomeration due to its better dispersion when compared to GO. The additional amount of CGO is up to 1.0 wt%, while that of GO is 0.75 wt%. These results are expected to provide a method for recycling chrome leather waste and for improving the performance of PVDF-based ultrafiltration membranes.

**Author Contributions:** Conceptualization, S.L. and H.H.; methodology, S.L. and H.H.; data curation, Y.H. and L.L.; writing-original draft preparation, Y.H. and X.W.; writing-review and editing, S.L. and Y.H.; project administration, S.L. and L.L.; funding acquisition, S.L. All authors have read and agreed to the published version of the manuscript.

**Funding:** This research was funded by National Natural Science Foundation of China (grant number 21276152) and Innovational Industrialization Foundation of Shaanxi Province of China (grant number 2016KTCL01-14).

**Conflicts of Interest:** We declare that we have no financial and personal relationships with other people or organizations that can inappropriately influence our work.

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
