# Peer review of "Enhanced Performance of PVDF Composite Ultrafiltration Membrane via Degradation of Collagen-Modified Graphene Oxide"

_applsci, doi:10.3390/app112311513_

Round 1
Reviewer 1 Report
Hou et al. reported that a recycled collagen-modified GO (CGO) could enhance the ultrafiltration membrane performance of PVDF-based films, in comparison to conventional GO. They claimed that the collagen recycled from chrome leather wastes could be grafted to the GO, resulting in better hydrophilicity. The increase in hydrophilicity of CGO minimized the agglomeration of the modified GOs into the PVDF matrix. To convince the hypothesis, the authors need to provide the surface properties of two model systems: (1) GO layer and (2) CGO-treated GO one, instead of the PVDF composites. The corresponding FT-IR, XRD, XPS results for the model systems will strongly support the hypothesis. In addition, two peaks at 2theta = 10.7 and 26 (weird) should be clarified in the XRD profiles of GO and CGO (Figure 2b). Sample preparation for TEM study was not described. Finally, this manuscript needs to be corrected in English.
Author Response
Dear Editor and Reviewers,
My manuscript, Enhanced performance of PVDF composite ultrafiltration membrane via degradation of collage- modified graphene oxide was revised according to reviewers’ comments. The itemized response to reviewer’s comments is attached. Changes to the manuscript are already marked in RED. Besides, the English language and style of the manuscript have also been edited by professional company. If there is any problem please let us know at once. Once again, thank you for your help to our paper processing.
For your guidance, itemized response to reviewer’s comments is appended below.
Hou et al. reported that a recycled collagen-modified GO (CGO) could enhance the ultrafiltration membrane performance of PVDF-based films, in comparison to conventional GO. They claimed that the collagen recycled from chrome leather wastes could be grafted to the GO, resulting in better hydrophilicity. The increase in hydrophilicity of CGO minimized the agglomeration of the modified GOs into the PVDF matrix. To convince the hypothesis, the authors need to provide the surface properties of two model systems: (1) GO layer and (2) CGO-treated GO one, instead of the PVDF composites. The corresponding FT-IR, XRD, XPS results for the model systems will strongly support the hypothesis. In addition, two peaks at 2theta = 10.7 and 26 (weird) should be clarified in the XRD profiles of GO and CGO (Figure 2b). Sample preparation for TEM study was not described. Finally, this manuscript needs to be corrected in English.
Response: Special thanks to you for your good comments. We have divided the above suggestions into four parts and responded to them one by one.
Point 1: the authors need to provide the surface properties of two model systems: (1) GO layer and (2) CGO-treated GO one, instead of the PVDF composites. The corresponding FT-IR, XRD, XPS results for the model systems will strongly support the hypothesis.
Response: Special thanks to you for your good comments. According to the suggestion, we pressed the GO and CGO samples into thin sheets and tested the contact angle to explain the improvement of CGO hydrophilicity. At the same time, the content about PVDF contact angle in the article is deleted.
Figure 2. (a) FTIR of GO and CGO (a); (b)XRD patterns of GO and CGO (b); (c)XPS spectra of GO; (d) XPS spectra of CGO; (e) Zeta potential of GO and CGO; and water contact angle of GO and CGO.
The results of FTIR, XRD and XPS indicated that the degraded collagen was grafted with GO successfully, and CGO contained more carboxyl groups. It can be seen from Figure 2e that the zeta potential of GO aqueous solution is -13.5 mv, and the CGO reaches -22.3 mv. Figure 2f shows that the contact angle of CGO was much lower than that of GO. Both showed that the carboxyl group of collagen gave CGO better hydrophilicity and increased its dispersion stability.
Point 2: In addition, two peaks at 2theta = 10.7 and 26 (weird) should be clarified in the XRD profiles of GO and CGO (Figure 2b).
Response: Special thanks to you for your good comments.
In order to check whether there are other problems in this result, we purified the two samples and re-conducted XRD test. The results are shown in the figure below. The results show that the absorption peaks still appear at these two positions. We intend to study the specific reasons for this result in the future.
Figure 2. (b)XRD patterns of GO and CGO
Point 3: Sample preparation for TEM study was not described.
Response: Special thanks to you for your good comments. The lack of sample preparation details in the article is due to the author's carelessness. According to the comment, we have added the sample preparation method to the article.
The sample was sliced, and transmission electron microscope (TEM) images were collected using a JEM-2010 (JEOL, Japan) at 200 kV.
Point 4: This manuscript needs to be corrected in English.
Response: Special thanks to you for your good comments. The English language and style of this article have been handled by a professional editing company.

Reviewer 2 Report
The authors descried fabrication of a new composite membrane for water filtration using PVDF, collagen, and graphene oxide. The results might include valuable new information. However, the manuscript was not well written. I request major revision as follows:
- Some parts of Abstract and Introduction are not convenient for readers. For example, definition of PVDF is described in Experimental section. However, this import word should be defined in Abstract and Introduction. The authors should carefully check such mistakes.
- I do not understand how to estimate porosity of the fabricate membranes. In the Experimental section, it is described that ‘The porosity is calculated according to the equation (1)’. However, There is no description how to obtain each parameter such as W1, W2, Vp, and Vm. Were those parameters obtained by some experiments? Or, was it a simulation study? The data in Figure 4 looks experimental data with error bars.
- Performance of the fabricate membrane (Figure 6) might be evaluated by some experiments. However, in the Experimental section, the authors just described some theoretical equations. There is no description of detailed experimental conditions. For example, how much the concentration of BSA? How much the pressure for the filtration? I think detailed experimental information is required.
Author Response
Dear Editor and Reviewers,
My manuscript, Enhanced performance of PVDF composite ultrafiltration membrane via degradation of collage- modified graphene oxide was revised according to reviewers’ comments. The itemized response to reviewer’s comments is attached. Changes to the manuscript are already marked in RED. Besides, the English language and style of the manuscript have also been edited by professional company. If there is any problem please let us know at once. Once again, thank you for your help to our paper processing.
For your guidance, itemized response to reviewer’s comments is appended below.
Point 1:.Some parts of Abstract and Introduction are not convenient for readers. For example, definition of PVDF is described in Experimental section. However, this import word should be defined in Abstract and Introduction. The authors should carefully check such mistakes.
Response: Special thanks to you for your good comments. This is caused by the author's carelessness. In view of this problem, the full text has been carefully checked and these errors have been corrected.
Point 2: I do not understand how to estimate porosity of the fabricate membranes. In the Experimental section, it is described that ‘The porosity is calculated according to the equation (1)’. However, There is no description how to obtain each parameter such as W1, W2, Vp, and Vm. Were those parameters obtained by some experiments? Or, was it a simulation study? The data in Figure 4 looks experimental data with error bars.
Response: Special thanks to you for your good comments.
Because we pay too much attention to the theoretical equation, it is inconsistent with the experiment. In this paper, we have modified the calculation formula according to the experiment.
Porosity is calculated according to the Equation (1).
|
, |
(1) |
Where ε is the membrane porosity (%), W1 is the dry membrane weight (g), W2 is the wet membrane weight (g), A is the membrane area (m2), l is the membrane thickness (m) and ρW is the water density (0.998 g·cm-3).
Point 3:Performance of the fabricate membrane (Figure 6) might be evaluated by some experiments. However, in the Experimental section, the authors just described some theoretical equations. There is no description of detailed experimental conditions. For example, how much the concentration of BSA? How much the pressure for the filtration? I think detailed experimental information is required.
Response: Special thanks to you for your good comments. According to comment, related content have been improved.
All experiments were conducted at 25 ℃ with a filtration pressure of 0.1MPa. For the measuring procedure, the pure water flux was recorded at 0.1MPa every 3 min for 90 min, and at least 4 replicated were collected to calculate an average value. The pure water flux of the membrane was calculated according to Equation (3).
|
, |
(3) |
Where Jw is the pure water flux (L·h-1·m-2), V is the volume of pure water passing through membrane per unit time (L), A is the membrane area (m2) and t is the filtration time (h).
The solute rejection performance of the membranes was measured by using BSA as foulants. The concentration of the feed and permeation was evaluated by a UV-vis double-beam spectrophotometer at a wavelength of 280 nm. The percentage of solute rejection was calculated by the Equation (4).
|
, |
(4) |
Where R is the BSA rejection rate of membrane (%), c1 is the concentration of the feed (1mg·mL-1), c2 is the concentration of permeation (mg·mL-1).

Round 2
Reviewer 2 Report
The manuscript has been well revised. Now It is suitable for publication.
Author Response
Thank you very much for your comment on our manuscript. We will continue efforts to do research project and write a better research paper.